# Challenges for *Cryptosporidium* Population Studies

**DOI:** 10.3390/genes12060894

**Published:** 2021-06-10

**Authors:** Rodrigo P. Baptista, Garrett W. Cooper, Jessica C. Kissinger

**Affiliations:** 1Center for Tropical and Emerging Global Diseases, University of Georgia, Athens, GA 30602, USA; jkissing@uga.edu; 2Institute of Bioinformatics, University of Georgia, Athens, GA 30602, USA; 3Department of Genetics, University of Georgia, Athens, GA 30602, USA; gwc32007@uga.edu

**Keywords:** cryptosporidiosis, mixed infections, population structure, genome evolution, molecular typing

## Abstract

Cryptosporidiosis is ranked sixth in the list of the most important food-borne parasites globally, and it is an important contributor to mortality in infants and the immunosuppressed. Recently, the number of genome sequences available for this parasite has increased drastically. The majority of the sequences are derived from population studies of *Cryptosporidium parvum* and *Cryptosporidium hominis*, the most important species causing disease in humans. Work with this parasite is challenging since it lacks an optimal, prolonged, in vitro culture system, which accurately reproduces the in vivo life cycle. This obstacle makes the cloning of isolates nearly impossible. Thus, patient isolates that are sequenced represent a population or, at times, mixed infections. Oocysts, the lifecycle stage currently used for sequencing, must be considered a population even if the sequence is derived from single-cell sequencing of a single oocyst because each oocyst contains four haploid meiotic progeny (sporozoites). Additionally, the community does not yet have a set of universal markers for strain typing that are distributed across all chromosomes. These variables pose challenges for population studies and require careful analyses to avoid biased interpretation. This review presents an overview of existing population studies, challenges, and potential solutions to facilitate future population analyses.

## 1. Introduction

Cryptosporidiosis is among the most important causes of diarrhea and diarrhea-associated death in young children in developing countries and is one of the major causes of waterborne outbreaks of illness in industrialized nations [1]. *Cryptosporidium* is an obligate, intracellular parasite that infects the epithelial cells of the digestive and respiratory tracts of a wide variety of hosts. It presents a significant public health problem, primarily for infants and the immunosuppressed [2]. There are 38 recognized *Cryptosporidium* species, which differ in host specificity and public health significance [3]. Of these, *C. parvum* and *C. hominis* are observed to be the most important sources of cryptosporidiosis in humans [3]. Additionally, there are over 20 species that have been identified molecularly as being responsible for zoonotic cryptosporidiosis in humans [4].

The *Cryptosporidium* life cycle alternates between asexual and sexual reproduction within a single host. Sexual recombination results in the production of oocysts and thus is essential to transmission, but recombination may also play a role in the continued infection of the host [5,6]. Mixed infections of *C. hominis* and *C. parvum* have been reported in human patients and are not uncommon in some areas [7,8,9,10].

Several different population structures have been reported for *Cryptosporidium* that vary both by species and context. Intra-species genetic structure diversity is observed in *Cryptosporidium* populations in endemic areas [11]. Populations have been observed in both panmictic [12] and largely clonal structures [13]. Some panmictic structures are masked by genetically identical clones at the loci examined (epidemic) [3,11,14]. In fact, the available studies focused on all these parasite population structures are usually observed in outbreaks and are limited by specific geographic location. The available data are often not comparable and can be affected by different transmission rates and animal husbandry practices. Different genes evolve at different rates. Thus, the selection of genetic markers for population studies depends on the biological question being addressed and the level of divergence that needs to be detected.

Population genetic studies based on highly polymorphic loci can shed light on the genetic diversity present within *Cryptosporidium*. Currently, however, only a few genetic markers have been developed for a few loci that do not cover all 8 chromosomes [15,16,17,18,19]. Whole genome sequence approaches are now being applied, when possible, to obtain a better understanding and overview of genetic variation and recombination within this species with the goal of better understanding its population structure [8,20,21] and evolution [22,23]. However, even with whole genome sequencing (WGS), challenges still remain for determining the global population structure of *C. parvum* and *C. hominis* as well as other human-infecting species. 

## 2. Current Status of *Cryptosporidium* Whole Genome Sequences

*Cryptosporidium* genome sizes are among the smallest reported in the Apicomplexa at ~9.1 Mb distributed in 8 chromosomes [24,25,26,27]. Analyses of genome content reveal broad-scale reduction and heavy reliance on transporters and host nutrients for survival. *Cryptosporidium* has very little intergenic content [25,26] and no reported mitochondrial or apicoplast genome sequences [28,29]. 

Currently, 52 genome assemblies are available for the genus *Cryptosporidium* in the NCBI GenBank [22,25,26,30,31,32,33], representing 15 species of the parasite (Table 1). Only *C. parvum* has chromosomal physical mapping information in support of the karyotype [27]. A few of the assemblies are annotated. The current reference sequences for each species have had to rely heavily on computational predictions and orthology to identify genes since little experimental expression evidence for genes exists, or existed, at the time of the annotation. 

The first assembled and annotated genome sequence was for *C. parvum* in 2004 [26]. This assembly used a HAPPY map approach combined with capillary sequencing. This assembly has had several annotation updates [34,35]. A newer genome assembly and annotation for *C. parvum*, which combines Illumina and long-read sequencing, has been generated. The new assembly, *C. parvum* (strain IOWA-ATCC), has no gaps, and all telomeres are identified [34]. *C. parvum* annotation updates over the years have identified new genes, corrected gene structures (mostly adding introns and untranslated regions, -UTRs), and identified non-coding genes [35,36]. There is a cluster of closely related species, including *C. parvum* and *C. hominis*, that share >95% nucleotide identity and high synteny relative to other species outside of this cluster [22,34]. The high sequence similarity within this group suggests that it is not inconceivable to develop markers that both cover and can disambiguate, multiple *Cryptosporidium* species.

## 3. *Cryptosporidium* Population Structure 

*Cryptosporidium* species lack variable morphological traits useful for identification. Single- [16,17,18,37,38,39,40,41,42,43] and multi-locus [19,44] typing tools and other approaches such as multiple-locus variable-number of tandem-repeats (VNTRs) [45,46] have been developed to help identify and characterize *Cryptosporidium* species and subtypes of this diverse genus. 

Cryptosporidia are primarily defined by host specificity and 18S ribosomal RNA sequence [18]. *Cryptosporidium* 18S rRNAs from different species differ by just a few nucleotides. However, some isolates with identical 18S rRNA sequences also present with different host specificities and phenotypes [22]. These observations revealed a subpopulation structure that was present within within the same species. Thus, the development of additional markers is needed to better understand *Cryptosporidium* population structure and evolution. 

Multi-locus typing of *C. parvum* revealed high genetic diversity within the species, including significant geographic segregation and complex population structure [11,47]. Analyses of population structure are an important guide to understanding transmission and evolution since extant organisms represent the outcome of their history and adaptation to their environment. The most commonly used genetic locus for subtyping *Cryptosporidium* spp. is the 60 kDa glycoprotein gene (*gp60*) [16]. This locus is useful because it contains multiple regions displaying high mutation rates, including, in particular, a “hyper-variable” microsatellite region. As previously reported, the number of *gp60* subtypes varies not just between species but also within them [47,48]. Nearly 20 different *gp60* subtypes have been observed in *C. parvum,* and they show a potential correlation to some observed phenotypes, e.g., subtypes IIa and IId, which are commonly found in zoonotic infections or a specific geographic location [4]. 

While single marker typing does reveal some correlation with host or phenotype in most studies, the results do not always agree with those of other genetic loci for some of the *C. parvum* subtype families, especially IIa and IId. In developing countries, IIa subtypes are rarely seen in humans, but in the middle east, both IIa and IId subtypes are commonly seen in humans. These observations show that strains carrying these *gp60* subtypes vary in phenotype and that using a single marker is still a low-resolution method to understand the genetic basis for host specificity or adaptation in *Cryptosporidium*. 

Several research groups have proposed and tested additional genetic markers, primarily based on microsatellites [14,49,50,51]. Most work well and reveal similar topologies within the same species and group of isolates used for development and testing. However, their performance declines when isolates from different geographic regions are analyzed [11]. An analysis of 11 different studies using several *loci* to type diverse isolates of *C. parvum* and *C. hominis* revealed that no single marker performs reliably, even in multi-locus studies (MLST), but *gp60* and TP14 show some promise in MLST with both species [19]. In the case of *C. parvum,* the MM19, MM18, MM5, MSF, MSD loci are identified as good options, and in *C. hominis,* the *cp47*, *msc6-7*, *rpgr*, ML2 loci worked well in the samples studied [19]. 

Fast evolving genes can be used to study the evolutionary dynamics of parasite population structure. Copy number variation (CNV) analysis of two fast-evolving protein families, MEDLE motif-containing proteins and insulinases-like proteases, have been linked to differences in host ranges between *C. parvum* and *C. hominis* [52]. A similar observation has been made within *C. parvum gp60* subtype families, where two sequenced isolates belonging to the IId subtype family, one from China and the other from Egypt, have lost one of the six genes encoding MEDLE proteins and gained at least one SKSR and insulinase-like protease gene when compared to the *C. parvum* IOWAII reference genome (subtype IIa) [53]. In addition to these gene gains and losses, *C. parvum* IIa, IId, and IIc subtype families have highly divergent subtelomeric genes encoding other families of secretory proteins [52], and as recently observed, there is subtelomeric genome plasticity in *C. parvum* [34].

A division of *C. parvum* into two branches that correspond to human-infecting (*C. parvum anthroponosum*) and non-human infecting (*C. parvum parvum*) has been suggested [22]. The authors examined 467 *gp60* sequences from 126 countries present in databases and 21 genome sequences (mostly from the UK). Analyses revealed evidence of positive selection and the existence of different population structures likely caused by different host migratory patterns. They also observed that the two branches had undergone genetic recombination, as evidenced by genetic exchanges between both branches and some incorporation of *C. hominis* related regions, mainly in *C. p. anthroponosum*. The majority of the detected recombination events are located in subtelomeric regions, which are a common hotspot for genes associated with host interactions and virulence in other apicomplexan parasites [54,55,56] and seem to play an important role in *Cryptosporidium*.

## 4. Challenges Faced by *Cryptosporidium* Population Studies

### 4.1. Sampling Limitations

There are numerous important studies of *Cryptosporidium* diversity in the literature. However, there is a size limitation in these studies. Most consist of a low number of isolates in a limited number of hosts and geographic locations [8,11,14,57,58]. As a consequence, the overall population structure of *C. parvum* and *C. hominis* remain unknown. A global study using a standardized, yet globally sensitive group of markers is still needed. 

### 4.2. Limited Number of Markers

18S ribosomal subunit markers are useful for the identification of species [59]. Subtypes of species, primarily within *C. parvum* and *C. hominis* can also be defined with additional markers, but some markers are more sensitive to local geographic diversity [47] and are not applicable globally. Many subtypes are only defined based on a few typing markers [60,61]. Single markers each have their own evolutionary history and are not always representative of the diversity of the genome sequence they are isolated from or indicative of the population structure of the organism being studied (Figure 1A). In fact, when performing an admixture analysis using all biallelic polymorphic sites detected in different *C. hominis* isolates available in NCBI GenBank [62], some *gp60* subtypes (loci) show incongruences in their ancestral subpopulation level (Figure 1B). A similar finding was observed for 46 *C. parvum* isolates in China, where the *gp60* subtypes showed population substructure variation according to Bayesian clustering of allelic data [11]. Early difficulties with obtaining genome sequence data from isolates has significantly hampered community efforts to develop a robust and universal set of markers that can be used to detect and compare the global diversity of extant isolates.

### 4.3. Mixed Infections

Individuals, especially in endemic regions, may be infected with multiple species [7] or, more commonly, multiple strains of *C. parvum* or *C. hominis.* Depending on the degree of relatedness of parasites within mixed infections and the markers used for detection, mixed infections might be missed [8,51]. Mixed infections can sometimes be detected with PCR [63], but more often, variants are detected with deep sequencing data [64,65,66]. Deep sequencing can also be used to estimate the rate of relatedness within the infection, but it will not yield complete genome sequences for each haplotype. 

The logical idea to clone individual parasites or oocysts of *Cryptosporidium* is, unfortunately, not an option since there is no optimal, prolonged, in vitro culture system, which can accurately reproduce the in vivo life cycle [67]. Some promising in vitro culture system models are emerging (see below, 6.1), and this prospect holds promise for teasing apart mixed infections and facilitating studies of recombination. Currently, available sequence data from isolates should be treated carefully as they may represent mixed infections [68]. 

Recently a study was able to generate single-cell sorting of oocysts for whole genome amplification, which indeed decreased the chances of obtaining information from multiple subtypes in a mixed infection [69]. *Cryptosporidium* has asexual and sexual components in its single-host life cycle, which enhances the chances of recombination. Indeed, recombination is reported in this parasite with linkage disequilibrium as short as 300 bp [8]. As a result, oocysts, which contain four meiotic haploid sporozoites inside, must be considered a population even if the sequence is derived from single-cell sequencing. These observed limitations may impact analyses of population structure in endemic areas of high transmission and high parasite diversity. 

### 4.4. Detection of Sexual Recombination in Cryptosporidium

*Cryptosporidium* has a single-host life cycle in which both asexual and sexual parasite stages are observed. Shed oocysts contain the meiotic progeny, sporozoites, and are proven to be the product of parasite sexual reproduction [6]. The presence of sexual reproduction in mixed infections of parasites increases the chances of finding recombinants that may pose a challenge for typing studies of populations as discussed above. 

Recombination has already been observed in *Cryptosporidium*. In *C. parvum*, recombinant progeny were detected from experimental mixed infections in INF-γ knockout mice [70]. Nader et al. have also shown that genetic introgression is detectable in the *C. parvum* genome and suggest that it may play a prominent role in its adaptive evolution and host-specificity [22]. 

In *C. hominis*, the use of several markers and whole genome sequencing has also permitted the detection of recombination. In this case, linkage disequilibrium was not observed. Evidence of recombination was detected at the 3’ end of Chr 1 and in three different regions of Chr 6 [20,52]. A study in Bangladesh was also able to find a decay of linkage disequilibrium between SNPs within 63 *C. hominis* isolates, evidencing recombination [8]. The diversity generated by recombinants could help to explain high rates of reinfection, seasonality, and differences in transmissibility, but proof awaits further study. 

The combination of mixed populations, recombination, an inability to clone and a lack of markers shows how complex and challenging it is to examine the population structure of *Cryptosporidium* and to identify potential hotspots of recombination if they exist. 

### 4.5. Lack of Metadata for Global Comparative Studies

Another major challenge for population analyses is the lack of metadata regarding sequenced isolates and a lack of required minimum information standards. Many groups are working with different local or regional sources of infection and usually collect metadata, when possible, that are useful for their studies but which may be incongruent with other studies. This problem is magnified by the fact that many isolates come from public health laboratories that do not have access to the relevant clinical or epidemiological data. Larger studies are needed to analyze existing data or collect prospect data. Additional metadata are needed and mechanisms should be utilized to preserve the subject’s right to privacy for examply by utilizing dbGaP. Metadata that would greatly facilitate the interpretation of genomic and population studies include: (i) characterized species (e.g., typed by a standardized marker, such as 18S SSU rRNA); (ii) sample type (oocyst, sporozoite, fecal DNA, etc.); (iii) geographic location of collection (with both country and city); (iv) date of collection; (v) collection source (environmental, host stool, culture system, etc.); (vi) *gp60* subtype; (vii) clinical severity; (viii) age; (ix) is this a repeat infection? (x) history of travel; (xi) source of water; (xii) possible zoonotic transfer and (xiii) association with a particular outbreak. Because of the metadata gaps that exist with currently available sequence data, many important observations and correlations cannot be determined, and the value of existing data for larger population studies is diminished. 

## 5. Detecting Mixed Populations in Collected Samples

Molecular characterization methods are usually the first step in the check for mixed infections. 18S/SSU rRNA and *gp60* are the most commonly used markers to identify *Cryptosporidium* species and genotype, respectively. Recently, a new tool called CryptoGenotyper was released. It shows 95.6% accuracy in detecting species in 18S/SSU rRNA data in mixed populations [71]. 

Mixed infections from subpopulations of the same species are complicated. Most molecular markers described are almost identical, sometimes differing only in single nucleotide variantion (SNVs). There are some bioinformatic approaches that can help detect the presence of mixed infections in WGS data. The ideal approach is to use variant detection software, such as GATK [72]. *Cryptosporidium* mixed infections tend to show multiple multi-allelic variants across all chromosomes, with clusters located in highly variable regions. As the genome sequence is haploid, alleles are not expected in clonal populations. While this approach can detect mixed infections, it cannot distinguish the different populations. 

The general term for identifying subsets from a mixed group is deconvolution. This process can be facilitated by having proportional mixed data in the sample input (e.g., frequencies of multiple genotypes with some divergence). Statistical in silico methods for deconvoluting multiple genome sequences present in an individual with mixed infections have also been developed for protozoan parasites present at unknown proportions. The DEploid package was developed specifically to deal with *Plasmodium* mixed infections [73]. It can estimate the number of strains and their relative proportions with some limitations. While promising, it is still unknown how well this approach will work with *Cryptosporidium*, since, unlike *Plasmodium*, which is asexual in human hosts, *Cryptosporidium* has sexual reproduction to generate oocysts within the host. Recombination events may impact the effectiveness of this approach.

## 6. Emerging Solutions to Deal with This Challenging Parasite

### 6.1. Promising In Vitro Cultivation Systems Parasites

Despite all the challenges, some solutions are arising from the community. Many promising in vitro culture systems are emerging for maintaining some species of *Cryptosporidium* parasites for an extended length of time. These include the hollow fiber cell culture system [74], three-dimensional and organoid tissue culture systems [75,76], and the air-liquid interface (ALI) cultivation system [77]. These systems are still new and present some limitations, especially the number of oocysts needed to seed the cultures. Optimization that would permit infections with a single oocyst or ideally with a single viable sporozoite (via cell sorting) would permit cloning as a routine methodology. The systems do not yet scale well, low numbers of parasites are obtained, and they require specialized equipment [78].

### 6.2. Sorted Single-Cell Genomic Sequencing

Mixed infections of different *Cryptosporidium* species and mixed subtypes of the same species occur in nature in the same host. Sexual reproduction also occurs within the same host, and recombinant progeny has been detected [6,7,10]. The extent to which different subtypes or even different species can have sex is currently unknown. If the complexity of mixed infections and the resulting mixed population of parasites can be reduced by cell sorting, this will greatly facilitate variant detection. Advances in the isolation of single oocysts and whole genome sequencing of *Cryptosporidium* from clinical samples are emerging. Single-cell sorting of oocysts for genomic analysis is a great solution that has enabled researchers to acquire and analyze genomic data from limited material [69]. Assays using single, sorted oocysts followed by whole genome amplification already show the great potential of this approach [69]. Importantly, single oocysts still need to be considered a population since they contain four haploid meiotic progeny (sporozoites). Using single-cell genome sequencing is a reliable way to examine and describe the genetic variation in complex populations, particularly low-frequency variation [79]. Unfortunately, because of the required amplification step, some analyses such as copy number variation (CNV) cannot be performed because they are biased by the amplification step.

### 6.3. Cryptosporidium Capture Enrichment Sequencing

Historically, the community has obtained DNA for sequencing using one of two approaches: (1) Antibody-based parasite oocyst capture from fecal or environmental samples, or, (2) propagation of oocysts in animal models (cattle, gnotobiotic piglet, and immunosuppressed mice) [31,80,81,82,83]. These approaches have the potential to restrict the levels of parasite diversity observed and thus impact our understanding of parasite diversity and biology. Antibodies have the potential to miss oocysts that do not bind well [80], and the passage of parasites obtained from one species in another, often unnatural host, can lead to selection. A critical assessment of the impact of these approaches will be fundamental to our understanding of *Cryptosporidium* biology and studies of its prevalence, virulence, diversity, and transmission. *Cryptosporidium* represents only a minute fraction of the fecal material and an even small fraction of the total fecal DNA. This fact impacts the sequencing costs and yield of parasite-specific DNA sequence. As a consequence, little is known about global *Cryptosporidium* genetic diversity because the numbers of oocysts collected are often small and previous technologies required too much DNA to permit proper whole genome characterization.

Capture Enrichment Sequencing (CES) is a target enrichment approach [84] that uses fairly long biotinylated single-stranded RNA baits (or probes) that are hybridized to complementary target DNA regions and are used to physically pull down the targeted DNA regions of interest for sequencing. This technique has been used with success for other apicomplexan parasites, such as *Plasmodium* in patient samples [85,86]. Since some genome sequences are available for different isolates and species of *Cryptosporidium*, they can be used to design specific bait-sets for the target and enrich these sequences. This approach has been piloted with *Cryptosporidium* with great success [87], and larger studies are underway [88]. The developed probe set will be made available to the community [89].

## 7. Conclusions

New methods are emerging to handle the numerous challenges that the *Cryptosporidium* community faces. Cryptosporidiosis mainly occurs in sporadic outbreaks and endemic settings, which suggests different evolutionary dynamics and population structures for each setting. Mixed infections and mixed or drifting populations as a result of recombination and replication errors combined with historical parasite culturing systems can and likely have impacted analyses and interpretations. Additional sequencing and global population structure analyses are needed to characterize extant diversity. As researchers continue to study outbreaks and additional geographic locations, markers capable of characterizing major population groups need to be developed to facilitate comparative analyses. Marker choices should be informed by the largest and most diverse set of sequences possible and should be distributed across all chromosomes. Microsatellites, by their nature, will have the greatest utility within local populations as they are unlikely to be universal enough to differentiate global diversity since they can arise easily. Having sensitive and reliable tools will be the key to better understanding *Cryptosporidium* biology and its transmission.

## Figures and Tables

**Figure 1 genes-12-00894-f001:**
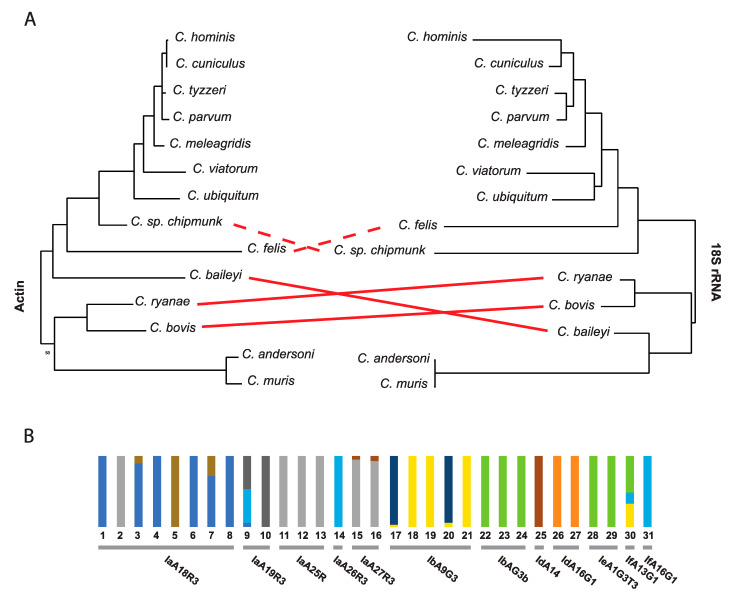
A single genetic marker is not representative of the entire genome sequence and evolution of an organism. (**A**) Comparative maximum likelihood topology analysis of two different *Cryptosporidium* genes (nt) that are usually used as markers; Dashed red lines represents differences with bootstrap values below 50% and solid lines with bootstrap values above 80% (**B**) Admixture clustering analysis of *C. hominis* biallelic variant sites reveals genomic variation within strains of the same *gp60* subtype. The number of ancestral populations (K) were predicted by the lowest cross-validation error K value (K = 10) obtained from the admixture analysis. Each column in the graph represents an individual isolate, while each color within the column represents an ancestral population. The *gp60* subtype for each isolate is indicated below the columns. GenBank and SRA accession numbers for the sequences utilized are provided in Appendix A and the methods are described in Appendix A.

**Table 1 genes-12-00894-t001:** Summary of *Cryptosporidium* genome assembly data available in the NCBI GenBank.

*Cryptosporidium* Species	# of Genome Sequences Available	Sequencing Technology	Gene Evidence Availability			
			RNAseq ^a^	Expressed Sequence Tag Datasets	Proteomic Data	# of Genome Annotations Available
*C. parvum*	19	Sanger, Illumina, 454, ABI SOLiD, PacBio, ONT, HAPPY map data	Yes	Yes	Yes	2
*C. hominis*	12	Sanger, Illumina, Ion Torrent, 454	Yes	Yes	Yes	5
*C. ubiquitum*	5	Illumina	No	No	No	1
*C. meleagridis*	3	Illumina	No	No	No	1
*C. andersoni*	3	Illumina	No	No	No	1
*C. muris*	1	Sanger and 454	No	No	Yes	1
*C. tyzzeri*	1	Illumina	No	No	No	1
*C. felis*	1	Illumina	No	No	No	1
*C. cuniculus*	1	Illumina	No	No	No	0
*C. ryanae*	1	Illumina	No	No	No	0
*C. bovis*	1	Illumina	No	No	No	0
*C. viatorum*	1	Illumina	No	No	No	0
*C. sp.* 37763	1	Illumina	No	No	No	0
*C. sp. chipmunk* LX-2015	1	Illumina	No	No	No	0
*C. baileyi*	1	Illumina, PacBio	Yes	Yes	No	0

^a^ not available for all lifecycle stages. Most data represent only extracellular oocyst and sporozoite lifecycle stages. **#** means Number.

## Data Availability

All data usedin this review was obtained from GenBank and all accession numbers are available in the Appendix A.

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
