# Peer review of "Challenges for Cryptosporidium Population Studies"

_genes, 2021, doi:10.3390/genes12060894_

Round 1

Reviewer 1 Report

  • Brief summary:

The review paper by Baptista, Copper and Kissinger deals with the complex issue of the population genetics of Cryptosporidium, and aims at highlighting the conceptual and practical challenges that have limited progress in this filed. It presents an updated information on the current whole genome data available and an historical account of the approaches used to characterize parasite isolates. It then clearly illustrates the importance of recombination in the evolution of these parasites, and the impact this process can have on the design of markers and interpretation of data. 

  • Broad comments highlighting areas of strength and weakness. These comments should be specific enough for authors to be able to respond.

The main strength of the review lies in the completeness of the presentation. All aspects are covered and the authors guide the reader through these aspects in a rationale way. The literature cited is appropriate and updated (see one minor remark).

There are, however, several areas of weakness.

First, as the issue they addressed is population genetics, I think they should provide some background and mention the population structures (panmictic, clonal, and epidemic) that a number of studies, properly cited in the reference list, already demonstrated to (co)exist in Cryptosporidium. This will allow showing how certain factors (e.g., transmission rate, husbandry practices) have an impact on the population structure.

Second, there is some confusion in the use of terms like gp60 genotypes and gp60 subtypes, or even gp60 subtype families. For non-experts, this terminology may be misleading. I suggest to clearly indicating in one introductory sentence what these terms are meant to indicate.

Third, the reader is awaiting for a discussion of the “potential solutions” (as it is claimed in the Abstract), at least from a theoretical point of view. Here the authors point at three things, namely new in vitro culture systems, single-cell genomic sequencing and capture enrichment sequencing. Regarding new in vitro culture systems, it is unclear how this can help to circumvent the problem of mixed populations. Regarding the capture enrichment, the way this is presented (and one can note that the listed references are abstracts of a Conference, therefore with little data included) seems to be applicable to mixed infections caused by different species. This is certainly of interest, but easier to tackle compared to situations where multiple genetic variants of the same species are present in the sample. Only the single-cell genomic sequencing represents a convincing solution; however, the necessary use of a whole genome amplification step, as correctly pointed out by the authors, is an inherent limitation. As a final remark, I noticed that the authors did not take a position regarding the nature of the markers (genes? repetitive sequences? intergenic sequences?) they consider appropriate for typing Cryptosporidium isolates at the global level.

  • Specific comments:

Line 95, IIt should read IId

Line 172-173: this sentence is important, but not so easy to interpret in its current wording. What the authors exactly mean by “the rate of relatedness in the infection”?

Relevant literature not cited

Grinberg A, Widmer G. Cryptosporidium within-host genetic diversity: systematic bibliographical search and narrative overview. International Journal for Parasitology 46 (2016) 465–471

Author Response

We thank Reviewer 1 for the helpful suggestions that improved this manuscript. Here are  the responses regarding the concerns presented:

First, as the issue they addressed is population genetics, I think they should provide some background and mention the population structures (panmictic, clonal, and epidemic) that a number of studies, properly cited in the reference list, already demonstrated to (co)exist in Cryptosporidium. This will allow showing how certain factors (e.g., transmission rate, husbandry practices) have an impact on the population structure.

R: Thank you for pointing this out. We agree that this should be addressed and a paragraph addressing different population structures has been added to section 1 of the review.

Second, there is some confusion in the use of terms like gp60 genotypes and gp60 subtypes, or even gp60 subtype families. For non-experts, this terminology may be misleading. I suggest to clearly indicating in one introductory sentence what these terms are meant to indicate.

R: Thank you for pointing this out, we made changes in the nomenclature to make it consistent as requested.

Third, the reader is waiting for a discussion of the “potential solutions” (as it is claimed in the Abstract), at least from a theoretical point of view. Here the authors point at three things, namely new in vitro culture systems, single-cell genomic sequencing and capture enrichment sequencing. Regarding new in vitro culture systems, it is unclear how this can help to circumvent the problem of mixed populations.

R: Thank you for the comment. We believe that having an optimized in vitro cultivation system would permit infections with a single oocyst or ideally even a single viable sporozoite (via cell sorting). A breakthrough ot this type would thus make cloning a viable methodology. This would in turn, help to deal with mixed infections. We modified the text on line 281-283 to make this perspective clearer.

Regarding the capture enrichment, the way this is presented (and one can note that the listed references are abstracts of a Conference, therefore with little data included) seems to be applicable to mixed infections caused by different species. This is certainly of interest, but easier to tackle compared to situations where multiple genetic variants of the same species are present in the sample.

R: We agree that it is still mainly focused between species, but the community is also working on intra species probes as in reference 91. We also clarified the text on line 304-311.  

As a final remark, I noticed that the authors did not take a position regarding the nature of the markers (genes? repetitive sequences? intergenic sequences?) they consider appropriate for typing Cryptosporidium isolates at the global level.

            R: We appreciate the observation made by the reviewer. We believe that it is still appropriate to develop a marker set for Cryptosporidium that will function at a global level. We added some sentences to clarify this position in section 7 lines 336-344.

  • Specific comments:

Line 95, IIt should read IId

R: Typo fixed

Line 172-173: this sentence is important, but not so easy to interpret in its current wording. What the authors exactly mean by “the rate of relatedness in the infection”?

            R: The text was modified at lines 187–192 to better clarify

Relevant literature not cited

Grinberg A, Widmer G. Cryptosporidium within-host genetic diversity: systematic bibliographical search and narrative overview. International Journal for Parasitology 46 (2016) 465–471

R: Thank you for pointing out this relevant review. Its citation was added to section 4.3. Line 199.

Reviewer 2 Report

The review deals with the challenges of molecular typing of Cryptosporidium for population studies.

The authors rightly claim that a sample with oocysts already constitute a “population” since even a single oocyst contains 4 sporozoites which are not clonal offspring because they are products of sexual reproduction and natural infections s are not derived from single cells but a number of oocysts which can create genetically diverse offspring.

Since precise determination of genotypes and subtypes is necessary to determine infection sources and to understand population structures (especially in the light of poor host specificity in some species and consequently high zoonotic potential), and, beyond that, can be used to determine virulence and other phenotypic, clinically relevant traits, the authors provide valid criticism and offer possible solutions for the problem of genotyping non-clonal eukaryotic pathogens.

The manuscript follows a clear structure and leads very nicely through the complex topic of genetics of cryptosporidia with a comprehensive overview of the currently available data.

Some suggestions for data collection and management (chapter 4.5) are not overly realistic (e.g. how can a zoonotic infection be determined in a parasite that is transmitted in many cases by water?), however, the underlying idea is appealing.

There are some minor points to be addressed in a revision that are listed below.

General remark: Sentences often start with the same, repetitive fill words, e.g. l. 73 and l. 77 “Cryptosporidium species” (2x) l. 104 “Several research groups” and l. 125 “some groups” as wells as l. 276 and l. 279: “Unfortunately” (2x). Some rewording would improve this.

  1. 125: no comma before “that”. A citation should be inserted.

l.127: Which is “This study”? There is no citation and this article is a review.

  1. 170: or, more commonly (insert comma after “or”)
  2. 1778: “some promising in vitro cultures…”: which ones? This should be outlined better and citations be added. Alternatively, the chapter 6.1. can be referenced to this previous text.
  3. 191: “Oocysts are meiotic spores”… what is meant by this? Oocysts themselves consist of protective layers covering first a zygote (before meiosis) and later (in Cryptosporidium) 4 unenveloped sporozoites which represent the actual cellular entities of the parasite. “Spore” is usually not an expression used in the context of apicomplexan biology.
  4. 197: Nader et al. have also shown (no comma and Nader et al implies a group of researchers, so plural).
  5. 282: Why is “Enrichment” is capital letters?
  6. 284: not cow but cattle (usually suckling calves)
  7. 288: “A critical assessment of the impact, or not...” what does this mean? Impact or lack of it?
  8. 304: a full stop is missing after the citation.

References: The format (font) must be adapted to match the rest of the manuscript. The citations, especially the journal abbreviations, must be checked for consistency. Doi numbers are with hyperlink, others without.

Author Response

We thank Reviewer 2 for the nice and important observations made. All suggested changes were made according to the reviewer comments.

General remark: Sentences often start with the same, repetitive fill words, e.g. l. 73 and l. 77 “Cryptosporidium species” (2x) l. 104 “Several research groups” and l. 125 “some groups” as wells as l. 276 and l. 279: “Unfortunately” (2x). Some rewording would improve this.

R: Fixed

  1. 125: no comma before “that”. A citation should be inserted.

R: Fixed

  1. 127: Which is “This study”? There is no citation and this article is a review.

R: Fixed

  1. 170: or, more commonly (insert comma after “or”)

R: Fixed

  1. 178: “some promising in vitro cultures…”: which ones? This should be outlined better and citations be added. Alternatively, the chapter 6.1. can be referenced to this previous text.

R: Fixed

  1. 191: “Oocysts are meiotic spores”… what is meant by this? Oocysts themselves consist of protective layers covering first a zygote (before meiosis) and later (in Cryptosporidium) 4 unenveloped sporozoites which represent the actual cellular entities of the parasite. “Spore” is usually not an expression used in the context of apicomplexan biology.

R: Fixed

  1. 197: Nader et al. have also shown (no comma and Nader et al implies a group of researchers, so plural).

R: Fixed

  1. 282: Why is “Enrichment” is capital letters?

R: Fixed

  1. 284: not cow but cattle (usually suckling calves)

R: Fixed

  1. 288: “A critical assessment of the impact, or not...” what does this mean? Impact or lack of it?

R: Fixed

  1. 304: a full stop is missing after the citation.

R: Fixed

References: The format (font) must be adapted to match the rest of the manuscript. The citations, especially the journal abbreviations, must be checked for consistency. Doi numbers are with hyperlink, others without.

R: Fixed